# Risk of acute kidney injury associated with anti-pseudomonal and anti-MRSA antibiotic strategies in critically ill patients

Jean-Maxime Côté[1,2,3]*, Michaël Desjardins[2,4,5], Jean-François Cailhier[1,2,6], Patrick T. Murray[3,7], William Beaubien Souligny[1,2]

1 Division of Nephrology, Centre hospitalier de l'Université de Montréal, Montréal, Canada, 2 Centre de recherche du centre hospitalier de l'Université de Montréal, Montréal, Canada, 3 Clinical Research Centre, University College Dublin, Dublin, Ireland, 4 Division of Infectious disease, Brigham and Women's Hospital, Boston, MA, United States of America, 5 Division of Microbiology and Infectious disease, Centre hospitalier de l'Université de Montréal, Montréal, Canada, 6 Institut du Cancer de Montréal, Montréal, Canada, 7 School of Medicine, University College Dublin, Dublin, Ireland

* Jean-maxime.cote@umontreal.ca

**Data Availability Statement:** The entire database used for this study (MIMIC-III) is publicly available following appropriate training from the MIT [https://mimic.mit.edu/]. Once permission has

## Abstract

### Background

An increased risk of acute kidney injury (AKI) with the widely prescribed piperacillin-tazobactam(PTZ)-vancomycin combination in hospitalized patients has recently been reported, but evidence in ICU patients remain uncertain. This study evaluates the association between the exposure of various broad-spectrum antibiotic regimens with *Pseudomonas* and/or methicillin-resistance *Staphylococcus aureus* (MRSA) coverage and the risk of AKI in critically ill patients.

### Methods and findings

A retrospective cohort study based on the publicly available MIMIC-III database reporting hospitalization data from ICU patients from a large academic medical center between 2001 and 2012. Adult patients receiving an anti-pseudomonal or an anti-MRSA agent in the ICU for more than 24-hours were included. Non-PTZ anti-pseudomonal agents were compared to PTZ; non-vancomycin agents covering MRSA were compared to vancomycin; and their combinations were compared to the PTZ-vancomycin combination. The primary outcome was defined as new or worsening AKI within 7 days of the antibiotic exposure using an adjusted binomial generalized estimating equation. Overall, 18 510 admissions from 15 673 individual patients, cumulating 169 966 days of antibiotherapy were included. When compared to PTZ, exposure to another anti-pseudomonal agent was associated with lower AKI risk (OR, 0.85; 95% CI, 0.80–0.91; p < .001). When compared to vancomycin, exposure to another anti-MRSA was also associated with lower AKI risk (OR, 0.71; 95% CI, 0.64–0.80; p < .001). Finally, when compared to the PTZ-vancomycin combination, exposure to another regimen with a similar coverage was associated with an even lower risk (OR, 0.63; 95% CI; 0.54–0.73; p < .001). A sensitivity analysis of patients with high illness severity showed similar results.

been granted by the MIT, anyone can access the entire MIMIC database to generate a new dataset (free of charge). All data used for this study can be obtained by contacting the Administrators of the MIMIC-III database (PhysioNet, from the MIT Laboratory for Computational Physiology) at contact@pgysionet.org. In addition, the entire dataset generated for this study can be shared upon request to the corresponding author once permission to access the original MIMIC database has been obtained from the MIT Administrators.

**Funding:** The author(s) received no specific funding for this work.

**Competing interests:** PTM previously received research funding from Abbott Laboratories and Alere Inc. and is receiving educational grant funding from Abbott Laboratories. He also received consulting fees from FAST biomedical, AM-Pharma, Renibus Therapeutics and Novartis. None of these disclosures are related to this work. The remaining authors have disclosed that they do not have any conflicts of interest.

## Conclusions

These results suggest that the risk of AKI in ICU patients requiring antibiotherapy may be partially mitigated by the choice of antibiotics administered. Further clinical trials are required to confirm these findings.

## Introduction

Antibiotics are widely used in intensive care units (ICUs) and help to save millions of lives. In a large international study, approximately 71% of ICU patients received antibiotics during their stay [1], representing a colossal number of patients worldwide exposed to antibiotics each year. Early empirical administration of broad-spectrum antibiotics is critical for the treatment of patients with severe infection. However, multidrug-resistant organisms are increasingly common and associated with a longer length-of-stay and higher mortality [2]. On the other side, inappropriate use of antibiotics is associated with the development of multidrug-resistant organism. The choice of the empirical antibiotic regimen should be individualised according to various factors such as local resistance rate, previous patient's infections as well as the suspected site of infection [3]. As infections caused by *Pseudomonas aeruginosa* and methicillin-resistant *Staphylococcus aureus* (MRSA) are becoming increasingly prevalent, the initial administration of broad-spectrum antibiotics regimen active against these organisms in high-risk patients has been endorsed by the Surviving Sepsis Campaign [4].

However, among the adverse effects of antimicrobial therapies, nephrotoxicity is a well-described complication for some of these antibiotic classes, such as aminoglycosides and vancomycin [5]. More recently, an increased risk of acute kidney injury (AKI) has been associated with the widely used piperacillin-tazobactam (PTZ) and vancomycin combination [6]. However, patients receiving this combination are likely to be severely ill–and consequently more at risk of AKI occurrence–than those who received other antibiotic classes. Moreover, recent groups have reported mixed results when analyzing the risk of AKI in critically ill patients receiving PTZ or PTZ-vancomycin when compared to carbapenems and carbapenem-vancomycin combination [6–8]. Furthermore, the severity of AKI is rarely reported. We have therefore hypothesized that this increased risk of AKI with the combination of vancomycin and PTZ might be limited to non-severe AKI and could be partially attributed to an inhibition of the tubular secretion of creatinine and might therefore represent *pseudo-nephrotoxicity* [6,8–10]. In this context, an increased risk of severe AKI should not be observed once adjusting for the severity of the illness. As a large proportion of ICU patients with a severe infection will receive an empirical anti-pseudomonal/anti-MRSA antibiotic regimen, confirming the safety of the PTZ-vancomycin is crucial.

This study, using the Medical Information Mart for Intensive Care (MIMIC)-III database, aimed to compare the risk of AKI in critically ill patients receiving PTZ and vancomycin, alone or in combination, to regimen with coverage for either *Pseudomonas*, MRSA, or both. We hypothesized a limited association with new of worsening AKI and no association with severe AKI requiring kidney replacement therapy (KRT) initiation.

## Methods

### Design and study population

This is a retrospective cohort study performed on the publicly available de-identified MIMIC-III database, derived from the Beth Israel Deaconess Medical Center's medical records,

Boston, MA. The database contains detailed information from 46,520 patients hospitalized between 2001 and 2012 [11]. The database was approved for research by the Massachusetts Institute of Technology (MIT) and the Beth Israel Deaconess Medical Center institutional review boards. Secondary use of this database does not require informed consent, which was waived from local ethic committee review.

Adult patients of 18 years and older receiving a broad-spectrum antibiotic of interest for ≥24 hours at the ICU were included. Patients with end-stage kidney disease (ESKD) at admission and patients who died within the first 24 hours of ICU admission were excluded. Multiple ICU admissions from the same patient were included if they all met eligibility criteria. This study followed the Strengthening the Reporting of Observational Studies in Epidemiology (STROBE) reporting guideline [12].

The following antibiotics of interest were categorized as follows: anti-pseudomonal (PTZ, ciprofloxacin, aminoglycosides [gentamycin, tobramycin, amikacin], ceftazidime, cefepime, carbapenems [imipenem cilastatin, meropenem] and aztreonam), anti-MRSA (intravenous vancomycin, daptomycin, linezolid) or anti-pseudomonal/anti-MRSA (combination of both classes).

## Observations and endpoints

The entire treatment duration for all antibiotics was converted into individual observation periods of 24-h (from 0:00 to 23:59). All periods where any antibiotics of interest were prescribed for at least 24-h (using pharmacy records data) were therefore included. For vancomycin, any additional observation period with a serum vancomycin level ≥10 mg/L were also included, even if no dose was administered at that day. The primary endpoint was new or worsening AKI within 7 days, defined as new-onset AKI or progression to a higher AKI stage for patients already having AKI criteria at the observation period, based on the serum creatinine KDIGO-AKI criteria (as depicted in S1 Fig, Suppl. Material) [13]. Secondary endpoints include new-onset KRT within 7 days and 30 days, a *proxy* for severe AKI. To explore the creatinine secretion inhibition attributed to PTZ-vancomycin, we compared the changes in serum creatinine and serum urea in the entire cohort, and patients with confirmed AKI (based on KDIGO-AKI staging).

## Covariates

Demographic characteristics include age, sex and ethnicity. Comorbidities were determined from the International Classification of Disease (ICD)-9 coding. Positive pressure ventilation was defined as receiving invasive or non-invasive mechanical ventilation documented by the presence of a recorded PEEP value on any blood gas result at the observation period. Leukopenia was defined as a white blood cell count <1.0 x10$^9$/mL, and corticosteroid as receiving any equivalent dose of prednisone ≥10mg/day during the antibiotic treatment duration. Active bacteremia was defined as any positive blood culture within three days of the observation period. No imputation was performed for these variables, except for the Sequential Organ Failure Assessment (SOFA) Score [14]. Due to multiple missing values, integration of the SOFA score in the multivariate model was limited. Therefore, missing values from each component of the score were imputed using the median.

## Statistical analysis

We used mean values (SD) for continuous variables and proportions for categorical variables for descriptive statistics. All statistical associations used PTZ as reference group for anti-pseudomonal agents, vancomycin for anti-MRSA agents, and PTZ-vancomycin for anti-

pseudomonal/anti-MRSA combinations. For the primary analysis, associations between the antibiotic regimen (exposure) and all endpoints were assessed using a generalized estimating equation (GEE) with a logistic (binomial) link function, producing Odds ratios (OR) with confidence intervals. This analysis considers the repeated measure design, implying that 24-h observation periods are not independent of each other and were clustered at the patients' level. An adjusted model that considered the most relevant factors, including non-traditional variables that showed a strong and clinically meaningful association with the risk of new or worsening AKI, was used for the multivariate model. Therefore, clinical endpoints were adjusted for age, sex, black ethnicity status, comorbidities (CKD, heart failure, liver disease and diabetes), SOFA score, hyperlactatemia, vasopressor requirement, positive pressure ventilation, bacteremia, corticosteroid therapy, leukopenia and antibiotic treatment duration.

Exploratory analyses included the following subgroup analyses: confirmed *Pseudomonas* spp. infection; confirmed MRSA infection; patients with CKD; antibiotics initiated within the first 48-hours of ICU admission; receiving the same antimicrobial agent for at least 72-hours; SOFA score >5 at ICU admission; excluding observations with known toxic vancomycin levels (>20 mg/L within 48h) and, finally, when excluding aminoglycoside exposure from the analysis. All primary analyses used a level of significance of 0.05. However, to compensate for multiple comparisons and potential type I error, results from all exploratory and subgroup analyses were interpreted as substantially significant only when $p<0.01$. Data source manipulation and variables selection were performed using the KNIME platform–The Konstanz Information Miner, version 4.3.0 (2021), while all statistical analyses were performed in R, version 4.0.3 (R Project for Statistical Computing) using the R-package "geepack" for general estimating equation [15] and SPSS 27.0 (Armonk, NY, IBM Corp).

## Results

### Patient selection and characteristics

Of the 46,520 patients reported in the entire MIMIC-III database, we identified 18,510 admissions from 15,673 eligible patients, totalizing 169,966 observation days (Fig 1). The mean (±SD) age was 65 (±16) year, and 43% were female (Table 1). Most patients (55%) were hospitalized in a cardiac or medical ICU, while vasopressors or positive pressure ventilation were required for at least one day during the antibiotic treatment duration in 31% and 34% of all admissions, respectively. An MRSA infection or colonization was confirmed in 1,945 (11%) admissions, while *Pseudomonas* spp. was identified in 1,263 (6.8%) of them. Most patients (74%) received broad-spectrum antibiotics within the first 48-hours of ICU admission. Regarding the choice of antibiotics, 6,136 (33%) of all admissions received PTZ for at least 24-hours, while 16,105 (87%) received at least one dose of intravenous vancomycin. The count of 24-h observations for each agent is further described in Table 2.

### Independent factors associated with the risk of AKI

Within the overall cohort, various factors such as heart failure, liver disease, and higher SOFA score at admission were associated with an increased risk of new or worsening AKI within 7 days (Fig 2). Similarly, active bacteremia, vasopressors, positive pressure ventilation requirement, and longer antibiotic duration were also associated with an increased risk of AKI progression at 7 days. In observations reporting exposure to vancomycin, higher serum levels were associated with a stepwise increase in the risk of AKI.

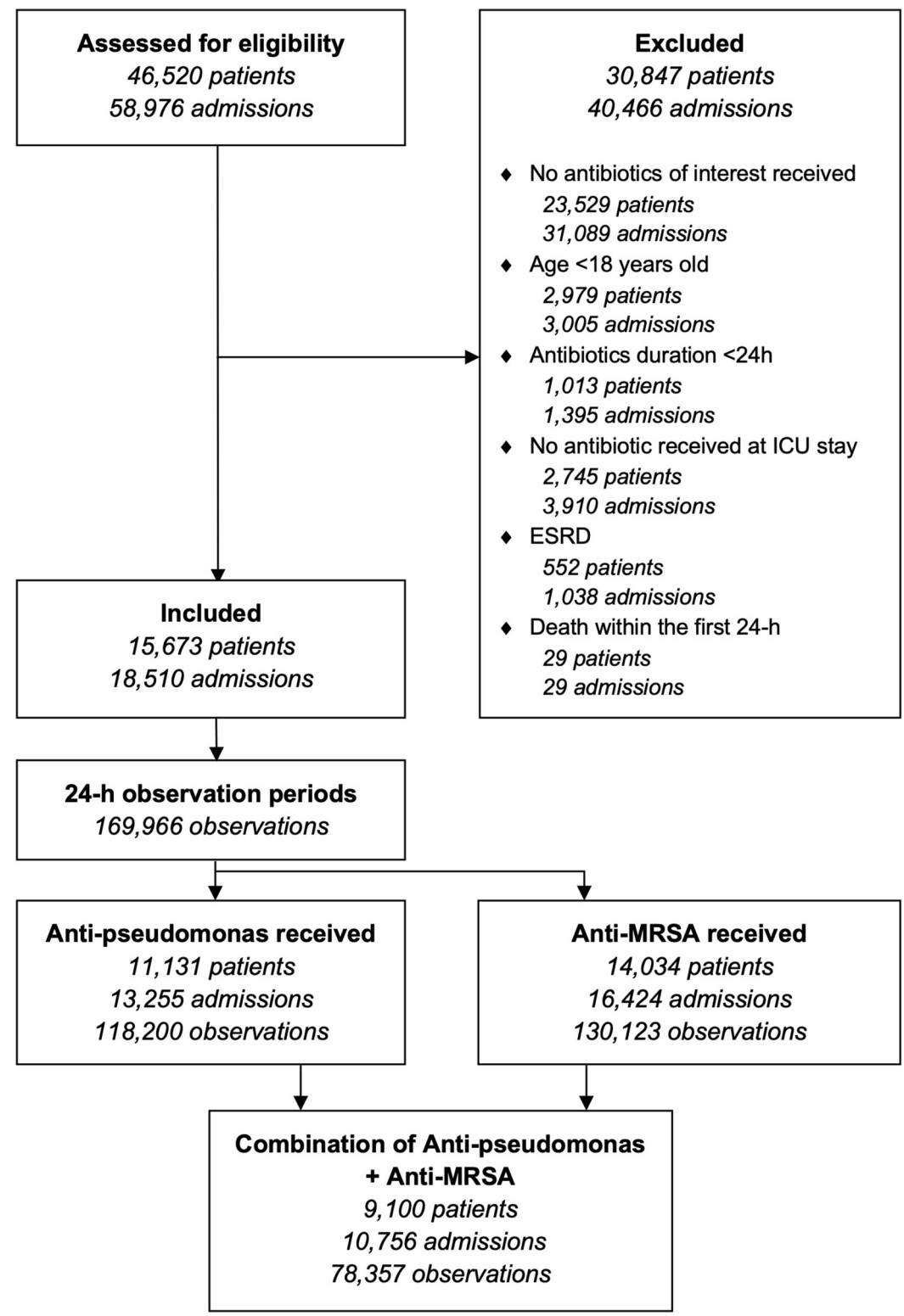

**Fig 1. Flowchart of included patients.**

**Table 1. Baseline characteristics of included patients[†].**

|  | All patients |
|---|---|
| ***Count, n*** | |
| Patients | 15,673 |
| Hospital admission | 18,510 |
| ***Demographics (%)*** | |
| Age, y (SD) | 65 (±16) |
| Male Sex | 10,468 (57) |
| Non-white ethnicity | 4,861 (26) |
| Diabetes | 5,780 (31) |
| Hypertension | 10,281 (56) |
| Heart failure | 6,175 (33) |
| Liver disease | 1,795 (9.7) |
| CKD, eGFR<60 mL/min[‡] | 3,867 (21) |
| eGFR<30 mL/min[‡] | 980 (5.3) |
| eGFR<15 mL/min[‡] | 218 (1.2) |
| ***Hospitalization and ICU data (%)*** | |
| MICU or CCU | 10,104 (55) |
| SICU or CSRU | 8,403 (45) |
| Positive pressure ventilation [¶] | 6,253 (34) |
| Vasopressor [¶] | 5,717 (31) |
| Lactate >2.4 mmol/L [¶] | 6,094 (33) |
| SOFA Score at admission (SD) | 6.3 (±3.9) |
| Hospital LOS, days (SD) | 14 (±13) |
| In-hospital mortality | 2,815 (15) |
| ***Infection-related factors (%)*** | |
| Leukopenia | 386 (2.1) |
| Corticosteroid [§] | 4,465 (24) |
| Confirmed *pseudomonas spp.* [¥] | 1,263 (6.8) |
| Confirmed MRSA [¥] | 1,945 (11) |
| Active bacteremia | 3,656 (20) |
| ***Antibiotics exposure (%)*** | |
| Started ≤48h of admission | 13,612 (74) |
| Piperacillin-tazobactam | 6,136 (33) |
| Ciprofloxacin | 5,633 (30) |
| Aminoglycoside | 1,590 (8.6) |
| Ceftazidime | 1,128 (6.1) |
| Cefepime | 3,389 (18) |
| Carbapenem | 2,394 (13) |
| Aztreonam | 541 (2.9) |
| Vancomycin | 16,105 (87) |
| Daptomycin | 474 (2.6) |
| Linezolid | 1,124 (6.1) |

[†] Based on hospital admission count.

[‡] Using the lowest creatinine value available within 3 months before admission (CKD-EPI).

[¶] At any time during the entire antibiotic treatment duration.

[§] At least 10 mg/day of prednisone (or equivalent) for at least one day of the antibiotic treatment duration.

[¥] Represent any positive Pseudomonas spp. and MRSA microbiology event, including colonization and active infection, within the same hospitalization.

**Table 2. Risk of new or worsening AKI and KRT associated with exposure to various anti-pseudomonas, anti-MRSA or their combination (multivariate).**

| | Observation days[†] | | AKI within 7d, OR [95% CI] | New onset KRT within 7d, OR [95% CI] | New onset KRT within 30d, OR [95% CI] |
|---|---|---|---|---|---|
| | Investiga-ted drug | Compa-rison | | | |
| **Non-PTZ anti-pseudomonas** (REF = PTZ) | 73,544 | 32,648 | 0.85 [0.80–0.91][***] | 0.86 [0.69–1.07][NS] | 0.72 [0.57–0.90][**] |
| **Non-vanco anti-MRSA** (REF = vancomycin) | 10,474 | 112,938 | 0.71 [0.64–0.80][***] | 1.13 [0.80–1.60][NS] | 0.58 [0.40–0.85][**] |
| **Non-PTZ anti-pseudomonas + non-vanco anti-MRSA** (REF = PTZ + vancomycin) | 6,471 | 22,873 | 0.63 [0.54–0.73][***] | 1.05 [0.61–1.79][NS] | 0.51 [0.26–1.01][NS] |

[NS]: p-value$\geq$.05, [*]: p-value < .05

[**]: p-value < .01

[***]: p-value < .001, AKI: Acute kidney injury, KRT: Kidney replacement therapy, REF: Reference group, PTZ: Piperacillin-tazobactam.

Results reported are Odds ratios with confidence intervals from a generalized estimating equation (binomial GEE) adjusted for: Age, sex, ethnicity, comorbidities (heart failure, liver disease and diabetes), SOFA score, hyperlactatemia, vasopressors, chronic kidney disease, antibiotic treatment duration, active bacteremia, positive ventilation, active corticosteroid therapy and leukopenia. Analyses for all anti-pseudomonal agents were also adjusted for the presence of a concomitant anti-MRSA agent, while analyses for anti-MRSA agents were adjusted for the presence of an anti-pseudomonal agent.

[†]Observations where both investigated, and comparator antibiotics were concomitantly received and where KRT was ongoing (ie. not at risk of progression) were excluded from the analysis.

## Association between the antibiotic class and the risk of new or worsening AKI

The primary outcome of new or worsening AKI within 7 days occurred in 7,578 (57%), 9,672 (59%) and 6,646 (62%) of admitted patients when exposed to anti-pseudomonal, anti-MRSA or their combination regimens, respectively. Stage 3 AKI or KRT initiation within 7 days occurred in 2,835 (21%) of patients exposed to anti-pseudomonas, in 3,064 (19%) exposed to anti-MRSA and in 2,586 (24%) exposed to both coverage for at least 24 hours (S1 and S2 Tables, Suppl. Material). In the multivariate analysis, when compared to PTZ, exposure to another anti-pseudomonal agent was associated with a lower risk of AKI within 7 days (OR, 0.85; 95% CI, 0.80–0.91; p < .001) (Table 2). When compared to vancomycin, exposure to another anti-MRSA agent was also associated with a lower risk of AKI within 7 days (OR, 0.71; 95% CI, 0.64–0.80; p < .001). Even lower risk of AKI was associated with exposure to the combination of a non-PTZ anti-pseudomonal and non-vancomycin anti-MRSA regimen (OR, 0.63; 95% CI, 0.54–0.73; p < .001). The associations between the risk of the primary outcome and exposure to each antibiotic agent solely or in combination are depicted in Table 3. These findings were also consistent in the univariate analysis and when considering at least 72h of treatment duration (S3 and S4 Tables, Suppl. Material).

Importantly, new or worsening AKI occurred within 7 days in 27.2%, 33.3% and 34.9% of observations exposed to PTZ only, to vancomycin only and to the PTZ-vancomycin combination respectively. Therefore, in the multivariate model, when comparing the PTZ-vancomycin combination to PTZ only, there was an increased risk of new or worsening AKI within 7 days (OR, 1.47; 95% CI, 1.36–1.60; p < .001). However, when comparing that combination to vancomycin only, no increased risk was observed (OR, 1.02; 95% CI, 0.96–1.08; p = NS) (Table 4).

## Association between the antibiotic class and new-onset of kidney replacement therapy

KRT occurred in 4% to 5% of all patients within 7 days and 30 days of antibiotic exposure for all three groups (S1 Table, Suppl. Material). As shown in Table 2, the association between the antibiotic regimen and the risk of KRT initiation within 7 days did not reach significance for

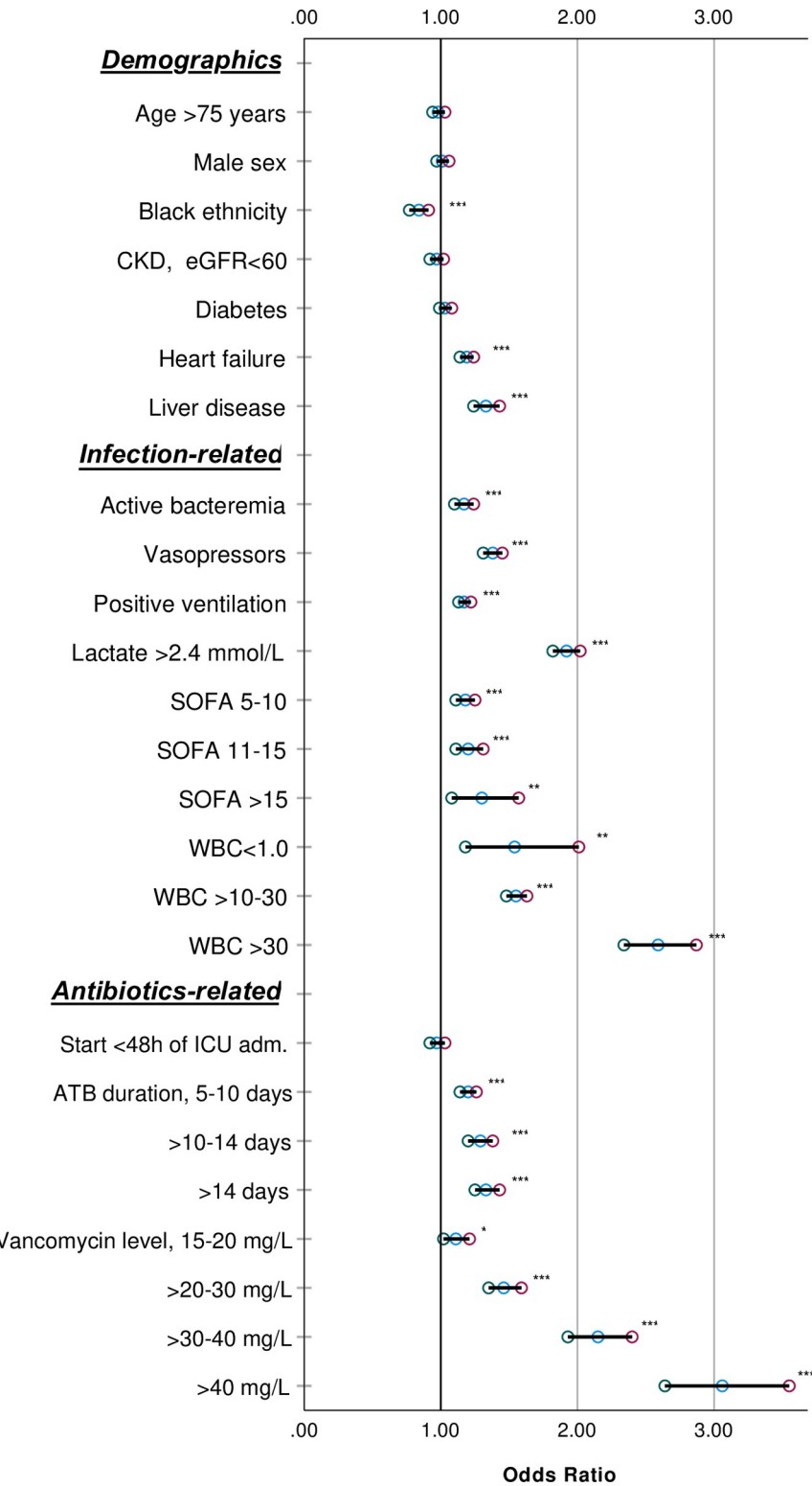

**Fig 2. Forest plot of major independent factors associated with new or worsening AKI (primary outcome) within 7 days (univariate).** Results reported are Odds ratios with confidence intervals from a generalized estimating equation (binomial GEE) non-ajusted. [NS]: p-value≥.05, [*]: p-value < .05, [**]: p-value < .01, [***]: p-value < .001.

**Table 3. Frequency of observations and risk of new or worsening AKI within 7 days associated with exposure to various anti-pseudomonas, anti-MRSA or their combination (multivariate).**

Frequencies of 24-hour observations reporting exposure to anti-Pseudomonas, anti-MRSA or both for the entire cohort, n (%)

| | | | Anti-MRSA | | |
|---|---|---|---|---|---|
| | | | Vancomycin | Daptomycin | Linezolid |
| | | | 118,909 (70) | 4,352 (2.6) | 8,696 (5.1) |
| Anti-pseudomonas | Piperacillin tazobactam | 41,265 (24) | 29,494 (17) | 599 (0.4) | 1,584 (0.9) |
| | Ciprofloxacin | 31,018 (18) | 13,572 (8.0) | 528 (0.3) | 773 (0.5) |
| | Aminoglycoside | 9,024 (5.3) | 4,876 (2.9) | 285 (0.2) | 579 (0.3) |
| | Ceftazidime | 7,319 (4.3) | 4,990 (2.9) | 88 (0.1) | 363 (0.2) |
| | Cefepime | 21,334 (13) | 15,865 (9.3) | 616 (0.4) | 702 (0.4) |
| | Carbapenem | 21,937 (13) | 11,993 (7.1) | 1,301 (0.8) | 2,317 (1.4) |
| | Aztreonam | 3,770 (2.2) | 2,679 (1.6) | 243 (0.1) | 333 (0.2) |

Risk of new or worsening AKI within 7 days associated with each individual antibiotic exposure, Odds ratio [95% CI]

| | | | Anti-MRSA | | |
|---|---|---|---|---|---|
| | | | Vancomycin | Daptomycin | Linezolid |
| | | | REF | 0.73 *** [0.61–0.87] | 0.72 *** [0.63–0.82] |
| Anti-pseudomonas | Piperacillin tazobactam | REF | REF | - | - |
| | Ciprofloxacin | 0.91 * [0.84–0.99] | - | 0.57 * [0.36–0.92] | 0.56 ** [0.38–0.84] |
| | Aminoglycoside | 1.30 *** [1.16–1.46] | - | 1.05 NS [0.57–1.91] | 0.90 NS [0.59–1.39] |
| | Ceftazidime | 0.83 ** [0.73–0.94] | - | 0.33 NS [0.07–1.63] | 0.74 NS [0.51–1.08] |
| | Cefepime | 0.91 * [0.84–0.99] | - | 0.62 * [0.40–0.97] | 0.66 * [0.43–0.99] |
| | Carbapenem | 0.67 *** [0.61–0.74] | - | 0.60 ** [0.44–0.81] | 0.73 * [0.57–0.93] |
| | Aztreonam | 0.65 *** [0.55–0.78] | - | 1.05 NS [0.59–1.88] | 0.25 *** [0.16–0.41] |
| | Anti-pseudomonas | | Anti-MRSA | | Anti-pseudomonas + Anti-MRSA |

NS: p-value≥.05

*: p-value < .05

**: p-value < .01

***: p-value < .001. REF: Reference group. MRSA: Methicillin-resistant staphylococcus aureus.

Results reported are Odds ratios with confidence intervals from a generalized estimating equation (GEE) adjusted for: Age, sex, ethnicity, comorbidities (heart failure, liver disease and diabetes), SOFA score, hyperlactatemia, vasopressors, chronic kidney disease, antibiotic treatment duration, active bacteremia, positive ventilation, active corticosteroid therapy, leukopenia. Analyses for all anti-pseudomonal agents were also adjusted for the presence of a concomitant anti-MRSA agent, while analyses for anti-MRSA agents were adjusted for the presence of an anti-pseudomonal agent.

Observations where both investigated, and reference antibiotics were concomitantly received and where KRT was ongoing (ie. not at risk of progression) were excluded from the analysis.

any of the three groups. However, when compared to PTZ, exposure to another anti-pseudomonal agent was associated with a lower risk of requiring KRT initiation within 30 days (OR, 0.72; 95% CI, 0.57–0.90; p < .01). Similarly, exposure to a non-vancomycin anti-MRSA agent was associated with a lower risk of KRT initiation (OR, 0.58; 95% CI, 0.40–0.85, p < .01). Despite a trend toward lower risk of KRT within 30 days, exposure to a non-PTZ or non-vancomycin combination did not reach statistical significance (OR, 0.51; 95% CI, 0.26–1.01; p = NS).

## Subgroup analyses

As shown in Fig 3 (and S5 Table, Suppl. Material), we found that associations between the risk of new or worsening AKI within 7 days and the choice of anti-pseudomonal agent, anti-MRSA

**Table 4. Risk of new or worsening AKI and KRT associated with exposure to the PTZ-vancomycin combination compared to PTZ or vancomycin individually (multivariate).**

| | Observation days[†] | | AKI within 7d, OR [95% CI] | New onset KRT within 7d, OR [95% CI] | New onset KRT within 30d, OR [95% CI] |
|---|---|---|---|---|---|
| | Investiga-ted drug | Compa-rison | | | |
| PTZ + Vancomycin (REF = Only PTZ) | 28,002 | 11,265 | 1.47 [1.36–1.60]*** | 1.17 [0.90–1.52][NS] | 1.28 [1.11–1.47]*** |
| PTZ + Vancomycin (REF = Only Vanco) | 28,002 | 86,390 | 1.02 [0.96–1.08][NS] | 1.30 [1.09–1.56]** | 1.14 [0.94–1.39][NS] |

[NS]: p-value≥.05

*: p-value < .05

**: p-value < .01

***: p-value < .001, AKI: Acute kidney injury, KRT: Kidney replacement therapy, REF: Reference group, PTZ: Piperacillin-tazobactam

Results reported are Odds ratios with confidence intervals from a generalized estimating equation (binomial GEE) adjusted for: Age, sex, ethnicity, comorbidities (heart failure, liver disease and diabetes), SOFA score, hyperlactatemia, vasopressors, chronic kidney disease, antibiotic treatment duration, active bacteremia, positive ventilation, active corticosteroid therapy and leukopenia.

[†]Observations where KRT was ongoing (ie. not at risk of progression) were excluded from the analysis.

agent or their combination were consistent among all subgroup analyses, except in patients with CKD or confirmed *Pseudomonas* infection.

## Exploration of the pseudo-nephrotoxicity

We compared the change in serum urea to the change in serum creatinine at 72-h of exposure to PTZ versus other anti-pseudomonal agents, with or without vancomycin (S6 Table, Suppl. Material). For the overall cohort, we found that exposure to PTZ was associated with an additional increase in serum creatinine by 3.2% (95% CI, 2.3–4.1%, p < .001), but not in serum urea (0.2%; 95% CI, ‾0.9–1.2%, p = NS) when compared to other anti-pseudomonal agents, with similar results when considering concomitant vancomycin administration. However, in patients who progressed to stage 2 or 3 AKI within 7 days, exposure to the PTZ-vancomycin combination was associated with a higher increase in serum creatinine than other anti-pseudomonas/anti-MRSA regimens (5.6% [95% CI, 3.1–8.2%, p<0.001]), with no significant difference in creatinine elevation in patients with limited stage 1 AKI. Instead, in that group with non-severe AKI, patients exposed to PTZ, with or without concomitant vancomycin, achieved the AKI creatinine elevation criteria despite a reduction in serum urea at 72-h.

## Discussion

To our knowledge, this is the most comprehensive ICU study reporting the association of AKI and the use of various anti-pseudomonal agents, anti-MRSA agents, and their combinations. We observed that both PTZ and vancomycin were associated with a higher risk of new or worsened AKI compared to other anti-pseudomonal or anti-MRSA agents, respectively. Moreover, there was a potential synergistic nephrotoxic association with the PTZ-vancomycin combination, where we showed an increased risk of new or worsened AKI when compared to PTZ only, as well as when compared to other regiments with *Pseudomonas* and MRSA coverage. Furthermore, these associations were robust in sub-group analyses.

An increasing body of observational cohort studies reported an increased risk of AKI with PTZ-vancomycin in hospitalized patients when compared to alternative broad-spectrum beta-lactams, such as carbapenems [6]. In high-risk patients, including those admitted to the ICU, this association was also reported as quite substantial, with Odds ratio up to 2.16 (95% CI,

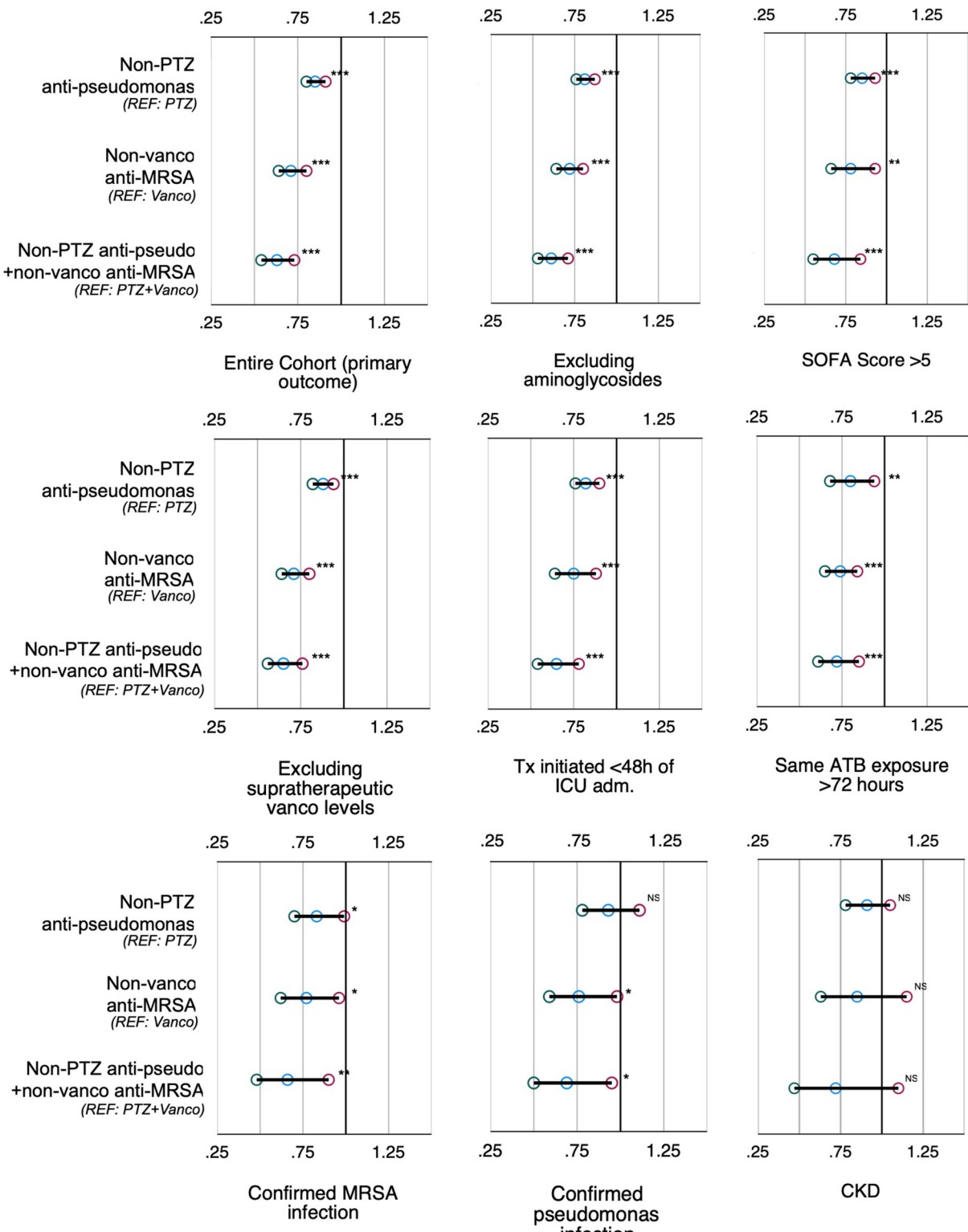

**Fig 3. Forest plot of the association between exposure to anti-pseudomonas, anti-MRSA and their combination with new or worsening AKI within 7 days for the entire cohort and all subgroup analyses (multivariate).** Results reported are Odds ratios with confidence intervals from a generalized estimating equation (binomial GEE) adjusted for: Age, sex, ethnicity, comorbidities (heart failure, liver disease and diabetes), SOFA score, hyperlactatemia, vasopressors, chronic kidney disease, antibiotic treatment duration, active bacteremia, positive ventilation, active corticosteroid therapy and leukopenia. Analyses for all anti-pseudomonal agents were also adjusted for the presence of a

concomitant anti-MRSA agent, while analyses for anti-MRSA agents were adjusted for the presence of an anti-pseudomonal agent. Using PTZ as reference group for all other anti-pseudomonal agents, vancomycin for all other anti-MRSA agents and the PTZ-vancomycin combination for all other regiment with anti-pseudomonal and anti-MRSA coverage. [NS]: p-value≥.05, [*]: p-value < .05, [**]: p-value < .01, [***]: p-value < .001, KRT: Kidney replacement therapy, REF: Reference group, PTZ: Piperacillin-tazobactam.

1.62–2.88) with the PTZ-vancomycin combination [16]. Another group using the WHO pharmacovigilance database reported a similar excess risk of AKI with the PTZ-vancomycin combination (ROR 2.1 [95% CI, 1.8–2.4]), that differed from other vancomycin-containing regimen [17]. In non-ICU studies, the increased risk of AKI attributed to the PTZ-vancomycin combination was mostly limited to non-severe AKI [18–20] and delayed AKI recovery [21]. Until now, most ICU studies were limited in size and none were powered to investigate an association with severe AKI or the receipt of KRT adequately.

Based on findings from these previous observational studies, our initial premise was an expected increased risk of AKI when exposed to PTZ or its combination with vancomycin, but limited to non-severe AKI [6]. Instead, we found a statistically significant increased risk of KRT initiation within 30 days associated with PTZ and vancomycin exposure, with a notable trend toward a similar risk with the PTZ-vancomycin combination. However, as with all observational studies, such findings should be interpreted cautiously as the reported association might be attributable to a residual confounding effect. We cannot extrapolate with certainty these findings resulting from secondary analyses, as many aspects of KRT prescription might have changed since 2001–2012, including the timing of initiation as recently investigated in the STARRT-AKI trial [22]. In addition, the proportion of patients who progressed to the primary endpoint of new or worsening AKI (57.0%) was slightly higher than usually reported in previous ICU cohorts [23], which might be due to the use of KDIGO criteria and selection of relatively sick patients with active infection requiring broad-spectrum antibiotics.

Our findings could not entirely corroborate the concept of *pseudo-nephrotoxicity* attributed to creatinine secretion inhibition with PTZ and/or vancomycin exposure. First, we showed an additional increase in serum creatinine but no change in serum urea at 72-h when considering the overall cohort as well as the subgroup of patients with severe AKI (S5 Table, Suppl. Material). We also showed, when exposed to PTZ as opposed to other regimens, that patients with non-severe AKI achieved the stage 1 AKI criteria (based on serum creatinine) despite a relative reduction in serum urea, which could be partially attributed to an inhibition of creatinine tubular secretion. However, a moderate elevation in serum creatinine solely due to its inhibition of tubular secretion is unlikely to translate into an increased risk of severe AKI and KRT use. Consequently, the contribution of this possible *pseudo-nephrotoxicity* is unlikely to explain the strong epidemiological association with severe AKI observed, as well as the presence of higher levels of kidney stress biomarkers previously reported [24,25].

Severe pseudomonas infections are associated with higher risk of therapeutic failure. Some guidelines have therefore recommended the use of double anti-pseudomonal coverage in high-risk patients, such as ventilator-associated pneumonia [3]. This study wasn't designed to evaluate the additional risk of AKI in patients requiring dual anti-pseudomonal coverage. However, as shown in Fig 3 (and S5 Table, Suppl. Material), among patients with confirmed pseudomonas infection, totalising 16,444 observation days, there was no increased risk of new or worsening AKI within 7 days associated with PTZ exposure. Therefore, in high-risk patients with known pseudomonas infection, PTZ may still represent an appropriate choice *a priori*.

Various pathophysiological mechanisms have been proposed regarding the PTZ-vancomycin-associated nephrotoxicity [6]. First, as other beta-lactams, PTZ may be associated with allergic acute interstitial nephritis, which could be aggravated in context of repeated exposure

in patients previously sensitized to PTZ [18,26]. Second, vancomycin was shown to induce production of reactive oxygen species and to increase mitochondrial and cellular stress [27], which might be exacerbated by an allergic interstitial nephritis (to beta-lactams) and other factors, including the infection itself. Vancomycin has also been associated with cast nephropathy is some reports [6]. However, no definitive mechanism to explain this synergistic toxicity has been confirmed, and histological correlation on kidney biopsies remains sparse [6].

Although we consider the question of PTZ-vancomycin associated nephrotoxicity to be a topical and clinically important issue, clinicians in ICUs have now several choices of antibiotics active against drug resistant gram-negative bacteria such as *Pseudomonas* spp., and gram-positive bacteria including MRSA. Adequate assessment of the overall risk of AKI attributed to the treatment received should be part of the equation when prescribing any therapy. The PTZ-vancomycin combination remains a pillar of broad spectrum antibiotherapy in critically ill patients. To our knowledge, no published randomized clinical trial has compared the most appropriate anti-pseudomonal/anti-MRSA regimen when considering efficacy and renal safety. However, patients still receiving the PTZ-vancomycin combination might benefit from additional kidney monitoring, including markers of kidney damage and careful dosing of vancomycin to achieve adequate but not excessive trough values.

Our study has several strengths. First, this is the largest cohort to investigate exposure to various anti-pseudomonal and anti-MRSA regimens and the risk of AKI in critically ill patients. In addition, to enabling extensive multivariate adjustment for comorbidities and severity of critical illness, we were also able to perform multiple subgroup analyzes to confirm the robustness of the observed associations. To leverage the large amount of data available, we used a longitudinal repeated measures approach, thereby avoiding a narrow focus on a specific time period during ICU stay and opting to include all periods of antibiotics exposure.

Limitations include potential ambiguity for comorbidities associated with ICD-9 codes. Also, data on medication administration were based on prescriptions rather than on administered drugs, and no dose-effect relationship on AKI risk could be evaluated with the PTZ exposure. Data come from a single center, and results may have been influenced by local practice in regard to antibiotic prescribing. The primary outcome measurement was based on serum creatinine variation only, using KDIGO-AKI criteria, which is sensitive to minor creatinine elevations and might have led to identification of AKI events with no clear clinical significance. Some have reported that the inclusion of urine output might improve AKI detection [28]. However, the pertinence of AKI defined solely by urine output criteria in the context of drug toxicity-associated AKI remains unknown. As with most retrospective studies, we cannot ascertain the clinician's intent underlying the use of these classes of antibiotics. Due to limited data, we cannot report the site of infection at the time of antibiotic administration, as well as previous exposure to these antibiotics or other beta-lactams. The total fluid balance could not be included due to the statistical design with repeated measures. Any positive blood culture was considered an active bacteremia as the MIMIC-III database did not allow to identify contaminant from true bacteremia. Finally, associations reported with secondary analyses should be interpreted cautiously as with any observational study.

## Conclusion

In critically ill patients who received anti-pseudomonal and anti-MRSA antibiotics, exposure to PTZ and vancomycin individually or in combination is associated with an increased risk of new or worsening AKI within one week. As this risk of AKI might be partially mitigated by the choice of antibiotics administered, clinicians should be aware of this added nephrotoxicity

when prescribing such combinations. Further clinical trials are required to confirm these findings prospectively.

## Supporting information

**S1 Fig. Example of the follow-up for the primary endpoint.**
(PDF)

**S1 Table. Absolute risk of major clinical endpoints for each antibiotic class received by admitted patients.**
(PDF)

**S2 Table. Absolute risk of major clinical endpoints for each antibiotic class received by observation periods.**
(PDF)

**S3 Table. Risk of new or worsening AKI and KRT associated with exposure to various anti-pseudomonas, anti-MRSA or their combination (univariate).**
(PDF)

**S4 Table. Risk of new or worsening AKI and KRT associated with exposure to various anti-pseudomonas, anti-MRSA or their combination (entire cohort and for at least 72h) (multivariate).**
(PDF)

**S5 Table. Risk of new or worsening AKI associated with exposure to various anti-pseudomonas, anti-MRSA or their combination (subgroup analyses, multivariate).**
(PDF)

**S6 Table. Change in serum creatinine and urea within 72-h associated with exposure to PTZ compared to another anti-pseudomonas with or without vancomycin.**
(PDF)

## Acknowledgments

All authors would like to thank the MIT for the management of the MIMIC database.

## Declarations

**Ethics approval and consent to participate.** The database was approved for research by the Massachusetts Institute of Technology (MIT) (No. 0403000206) and the Beth Israel Deaconess Medical Center (No. 2001-P-001699/14) institutional review boards. Secondary use of this database does not require informed consent, which was waived from local ethic committee review.

## Author Contributions

**Conceptualization:** Jean-Maxime Côté, Michaël Desjardins, Jean-François Cailhier, Patrick T. Murray, William Beaubien Souligny.

**Data curation:** Jean-Maxime Côté.

**Formal analysis:** Jean-Maxime Côté.

**Methodology:** Jean-Maxime Côté.

**Software:** Jean-Maxime Côté.

**Supervision:** Patrick T. Murray, William Beaubien Souligny.

**Validation:** William Beaubien Souligny.

**Writing – original draft:** Jean-Maxime Côté.

**Writing – review & editing:** Michaël Desjardins, Jean-François Cailhier, Patrick T. Murray, William Beaubien Souligny.

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
