## [Decision Letter · Decision Letter 0]

4 Jan 2022

PONE-D-21-34271Risk of Acute Kidney Injury associated with Anti-pseudomonal and Anti-MRSA Antibiotic Strategies in Critically ill PatientsPLOS ONE

Dear Dr. Cote,

Thank you for submitting your manuscript to PLOS ONE. After careful consideration, we feel that it has merit but does not fully meet PLOS ONE’s publication criteria as it currently stands. Therefore, we invite you to submit a revised version of the manuscript that addresses the points raised during the review process.

We look forward to receiving your revised manuscript.

Kind regards,

Eili Y. Klein, PhD

Academic Editor

PLOS ONE

Journal Requirements:

Additional Editor Comments (if provided):

Sorry for the delay in review. Given the ongoing pandemic I allowed extra time for the reviewers to complete their reviews. That extra time allowed for four reviews. There was a split on the outcome, but I think all agreed that it is an important and useful paper. While the attention to the statistical associations is appreciated, the question was raised by one of the reviewers why propensity scoring of some type was not included as part of the analysis given its wide-spread use in other AKI analyses. Given the uncertainty about the underlying reasons driving prescribing, on could argue that the groups in the paper may not be fully comparable (thus the reason at least one reviewer mentioned the need to discuss an RCT). Adding a propensity scoring algorithm of some flavor would significantly strengthen the paper and tie it to the broad literature about AKI already. Additionally, the question of why the PTZ-vancomycin combination was more nephrotoxic than PTZ alone but not any more nephrotoxic than vanc should be discussed further as mentioned by the reviewers.

Reviewers' comments:

Reviewer's Responses to Questions

**Comments to the Author**

1. Is the manuscript technically sound, and do the data support the conclusions?

Reviewer #1: Yes

Reviewer #2: Yes

Reviewer #3: Yes

Reviewer #4: Partly

2. Has the statistical analysis been performed appropriately and rigorously? 

Reviewer #1: Yes

Reviewer #2: Yes

Reviewer #3: Yes

Reviewer #4: Yes

3. Have the authors made all data underlying the findings in their manuscript fully available?

Reviewer #1: Yes

Reviewer #2: Yes

Reviewer #3: Yes

Reviewer #4: Yes

4. Is the manuscript presented in an intelligible fashion and written in standard English?

Reviewer #1: Yes

Reviewer #2: Yes

Reviewer #3: Yes

Reviewer #4: Yes

5. Review Comments to the Author

Reviewer #1: Thank you for doing this study and asking me to review. This issue has been topical in recent years and there are several unanswered questions. This article contributes to our understanding of nephrotoxicity with vancomycin and piperacillin-tazobactam in critically ill patients, particularly due to the large number of cases included. I am sure the authors will agree with me that it a prospective RCT is needed to properly answer the question.

My general thoughts about the paper are:

Teasing out a single cause of nephrotoxicity in critically ill patients is hard with multiple confounders

This is demonstrated by Figure S1: 18 of 25 exposures assessed were associated with increased rate of kidney injury

Patients needed only to have 24 hours of antibiotics to be included in this study. I am skeptical that acute kidney injury could be blamed on these antibiotics for such a short timeframe. Admittedly, had the authors chosen a different timeframe their results would have shown higher levels of AKI in these antibiotic groups. I also acknowledge that the duration of Piptaz/Vanco/combination needed to cause AKI is not known

Progressive increase in rates of kidney injury with longer duration of antibiotic therapy is important. As with all elements of this study, the finding has many confounders but I still think it is an interesting and relevant finding.

It is unusual for patients who are on treatment for both P. aeruginosa and MRSA to not be prescribed either Piptaz or vancomycin. In my experience, if there is a concern re nephrotoxicity with combination vancomycin and piperacillin/tazobactam, patients are usually treated with cefepime or meropenem instead of Piptaz. I would be more interested in rates of AKI with cefepime/vancomycin or meropenem/vancomycin compared to Piptaz/vancomycin.

I like the included tables. However, I feel some of the most useful data is buried in the supplementary section, particularly Figure S1 and Tables S1 and S2.

The discussion makes a good point about changes in indications for dialysis in ICU over the study timeframe.

Completely agree that the findings of this study add yet more weight to the requirement for prospective RCTs on this issue rather than further retrospective data

Reviewer #2: Thank you for the opportunity to review, "Risk of Acute Kidney Injury associated with Anti-pseudomonal and Anti-MRSA Antibiotic Strategies in Critically ill Patients."

This is a retrospective cohort study of data from the MIMIC-III database including 18,510 encounters among 15,673 individual adult patients admitted to a single academic center’s ICUs over an 11-year period (2001-2012). The authors analyze the association between treatment with Vancomycin vs another anti-MRSA antibiotic, Piperacillin-Tazobactam (PTZ) vs another anti-Pseudomonal antibiotic, and combination therapy with Vancomycin-PTZ vs any other anti-MRSA-non-PTZ anti-Pseudomonal antibiotic combination with new or worsening acute kidney injury (AKI) within 7 days, according to creatinine-based KDIGO criteria. Secondarily, the authors analyze the association between the above antibiotics and new-onset renal replacement therapy within 7- and 30-days and changes in serum creatinine and serum urea among all study participants. The authors conclude that among ICU patients who received anti-Pseudomonal and anti-MRSA antibiotics, exposure to PTZ and vancomycin, individually or in combination, had an increased risk of new or worsening AKI within one week.

This is a well-performed research study that follows the STROBE criteria and adds to an existing body of evidence regarding the renal risks of exposure to various nephrotoxic medications in critically ill patients.

Major revision suggestion:

My only major criticism is the authors’ decision to not perform propensity score matching in their retrospective analysis of this observational dataset. Many recent well-performed large-scale retrospective cohort studies seeking to elucidate the association between various interventions and acute kidney injury have used this statistical tool to good effect and have been published in PLoS One (PMID 30142190) and other journals (PMID 31952871). The authors should consider whether this would strengthen their analysis by accounting for the numerous patient characteristics that may contribute to clinicians’ decision to choose one antibiotic agent over another in critically ill patients.

Minor revision suggestions include:

- Page 5, line 11: Missing closing parenthesis after “aztreonam”

- Page 14, line 1: Replace “but” with “except in” to make it clearer that these two sub-groups had different outcomes measured. As currently written, this is unclear.

- Page 14, Discussion section: The authors might consider discussing whether their sub-group analysis demonstrating that, among patients with confirmed Pseudomonal infection, administration of PTZ was not associated with worsening AKI might encourage clinicians to more carefully consider which patients are at higher risk for Pseudomonas to help determine when PTZ may indeed be the optimal choice of anti-Pseudomonal antibiotic if it does not have a significant association with new or worsening AKI for patients who end up having a Pseudomonal infection

- Page 17, line 17-19: Are the authors able to determine how often their studied antibiotics were prescribed but not administered? One might assume this would be a very small number and, if the authors are able to report how frequently this occurred, the reader would be better able to determine how much this inherent limitation of a database study should inform their conclusions.

Reviewer #3: This article examines the association of piperacillin-tazobactam with acute kidney injury (AKI) in a large single-center cohort of ICU patients. As stated by the authors, the concept is not new but the current study provides a more granular analysis of this association by controlling for common risk factors for AKI n ICU patients. Few comments and queries, if I may.

1. The finding that PTZ-vancomycin combination was more nephrotoxic than PTZ alone but not any more nephrotoxic than vanc alone is interesting and should be discussed further. How do the authors explain this finding? Does this suggest that when patients develop AKI while on this combination, the relative contribution of vancomycin is higher than PTZ? Does this finding suggest that for patients who--- for whatever reason--- are thought to require vancomycin as a preferred agent, the addition of PTZ should not expose them to higher risk of nephrotoxicity than vanc alone? Similarly, for those who require PTZ as the drug of choice (eg, when risk of cefepime-induced encephalopathy is considerable), the addition of vanc should be avoided unless it also is considered to be the drug of choice?

2. Is there data examining 24, 48 or 72 hrs of the combination vs longer duration as relates to risk of AKI? Please discuss how the results of this study may be used (or cannot be used) to determine if there is a “safe” duration of PTZ-vancomycin combination beyond which the risk of AKI increases significantly.

Others…

1. Abstract, line 17-18. Should be a complete sentence.

2. Page 5, line 16. Should be “was”

3. Page 6, line 13. Active bacteremia was defined as “any positive blood culture…” How were contaminants handled? Consider using the term “true bacteremia” with explanation of excluding contaminants based on clinical grounds.

4. Page 11, Table 2. Consider an alternative to “Intervention” column heading since this is a retrospective (not prospective) study. ? Investigated drugs

Thank you!

Reviewer #4: The manuscript examined the risk of AKI associated with anti-pseudomonal and anti-MRSA antibiotic strategies in critically ill patients. Here are some comments for consideration:

1. Page 3, line 6. May want to further expand on relationship between broad spectrum antibiotics and MDR organisms.

2. Page 3, line 11. Would want to update reference to the new Campaign recommendations for 2021.

3. Page 3, line 21. The study by Blevins A. AAC; 2019; 63: e02658.may be useful to include as that study examined the rates of AKI with different severity in the ICU. The study by Contejean A. JAC 2021; 76: 1311 would also be useful in the discussion on AKI with vancomycin/piperacillin/tazobactam combination.

4. Page 5, line 2. Is there a specific time frame for overlap of antibiotics?

5. Page 5, Observations and endpoints. Would be useful to define severe and non-severe AKI as this was discussed as a limitations of previous studies.

6. Page 5, line 21. Can further clarify day 7 observation period, is that after receipt of combination antibiotics or any antibiotics?

7. Page 6, covariates. Was patients in shock included in the study? If so, where they on vasopressors as this was considered a major risk factor for Aki development.

8. Page 7, line 10. Were comorbidities like shock, other nephrotoxins, and amount of fluids administered included in the covariate?

9. Page 7, line 17. What is defined as toxic vancomycin level?

10. Page 8, table 1. Is there a breakdown of what stage of CKD for the patient population?

11. Page 8, table 1. Is the mean time to treatment and mean time to AKI available?

12. Page 10. Is the data for AKI by severity available?

13. Page 14, line 6. This is referring to Table S5 rather than S6.

14. Page 14, line 11. Can expand on this statement a little more.

15. Page 14, discussion. Can discuss more on the overall incidence rate of AKI in this study compare to other studies. Can also discuss about the difference in outcome when observing the rates of AKI in patient receiving piperacillin vs those who received vancomycin.

16. Page 17, line 21. Can comment more about the limitation of using KDIGO-AKI criteria vs. the other definitions.

6. PLOS authors have the option to publish the peer review history of their article (what does this mean?). If published, this will include your full peer review and any attached files.

Reviewer #1: No

Reviewer #2: No

Reviewer #3: No

Reviewer #4: No

---

## [Author Response · Author response to Decision Letter 0]

18 Jan 2022

Editor Comments: 

Q: Sorry for the delay in review. Given the ongoing pandemic I allowed extra time for the reviewers to complete their reviews. That extra time allowed for four reviews. There was a split on the outcome, but I think all agreed that it is an important and useful paper. While the attention to the statistical associations is appreciated, the question was raised by one of the reviewers why propensity scoring of some type was not included as part of the analysis given its wide-spread use in other AKI analyses. Given the uncertainty about the underlying reasons driving prescribing, on could argue that the groups in the paper may not be fully comparable (thus the reason at least one reviewer mentioned the need to discuss an RCT). Adding a propensity scoring algorithm of some flavor would significantly strengthen the paper and tie it to the broad literature about AKI already. Additionally, the question of why the PTZ-vancomycin combination was more nephrotoxic than PTZ alone but not any more nephrotoxic than vanc should be discussed further as mentioned by the reviewers.

R: First of all, thank you very much for accepting to review our work. We consider the antibiotics’ associated nephrotoxicity as a relevant clinic topic and data regarding ICU were lacking. We agree with you that Propensity Score (PS) Methods are increasingly used in large observational studies as a useful alternative to more conventional co-variate adjustment. Propensity score (especially methods using PS matching or PS stratification) is a reliable method to provide co-variate balance and is relatively easy to interpret. However, their superiority to standard co-variate adjustment in large dataset remains unclear1. 

In our study, a generalized estimating equation (GEE) model has been performed, to allow the use of a repeated measures design clustered at the patient’ level. It means that a single patient, during his entire ICU stay, could have been exposed to multiple antibiotic combinations successively and been therefore classified with multiple exposure at different timepoints. For example, the same patient can be part of the ceftazidime-vancomycin “group” for some 3 observation days, part of the meropenem-vancomycin “group” for 4 other observation days, and finally only received meropenem for the last 2 days of observation. As result, including a patient-level co-variate(s) propensity score matching does not seem methodologically appropriate.

The only PS method that could be integrated into such GEE analysis would be computing a PS for all timepoints included (not per patient) and integrate it as a covariable into the final adjustment model. However, as shown par Elze et al.1, adding the PS as an additional covariate in large dataset (like our study) produces results very similar to standard co-variate adjustment, with similar estimates, SEs and no clear statistical benefits, especially when the high number of events does not limit the number of covariable than can be integrated into the models (as this is the case here). We therefore consider that our primary multivariate model using 15 different co-variates with clinical and strong statistical relevance (based on Table S1, Suppl Material)(e.g. age, sex, ethnicity, CKD, heart failure, liver disease, diabetes, SOFA score, lactatemia, vasopressor requirement, ventilation, bacteremia, corticosteroid therapy, leukopenia and antibiotic treatment duration) was enough to compensate for potential confounders. 

1 Elze, M.C. et al. J Am Coll Cardiol. 2017;69(3):345-57 

Reviewers' comments:

Reviewer's Responses to Questions

Comments to the Author

 Review Comments to the Author

Reviewer #1: 

Q: Thank you for doing this study and asking me to review. This issue has been topical in recent years and there are several unanswered questions. This article contributes to our understanding of nephrotoxicity with vancomycin and piperacillin-tazobactam in critically ill patients, particularly due to the large number of cases included. I am sure the authors will agree with me that it a prospective RCT is needed to properly answer the question.

R: Thank you for reviewing our manuscript. Indeed, we certainly agree with you that prospective trials are lacking. 

Q: My general thoughts about the paper are:

Teasing out a single cause of nephrotoxicity in critically ill patients is hard with multiple confounders

This is demonstrated by Figure S1: 18 of 25 exposures assessed were associated with increased rate of kidney injury

Patients needed only to have 24 hours of antibiotics to be included in this study. I am skeptical that acute kidney injury could be blamed on these antibiotics for such a short timeframe. Admittedly, had the authors chosen a different timeframe their results would have shown higher levels of AKI in these antibiotic groups. I also acknowledge that the duration of Piptaz/Vanco/combination needed to cause AKI is not known

Progressive increase in rates of kidney injury with longer duration of antibiotic therapy is important. As with all elements of this study, the finding has many confounders but I still think it is an interesting and relevant finding.

R: We agree that 24h is a short exposure to develop associated nephrotoxicity. However, it should be noted that a GEE (with a repeated measures design, using 24h observations) has been used to compare each antibiotic exposition. It means that, with this analysis, the effect of an exposure (ATB X) on the observed result (rate of AKI) is relatively proportional to the time of the exposure. For example, a patient exposed 7 days to piperacillin will add 7 observations to the “cohort”, while a patient exposed to 24h will add only 1 observation. Therefore, in this GEE the chance of measuring an effect of the exposure on the outcome is proportional to the duration of the treatment receive. Also, allowing short exposure (as 24h observation) has the advantage to identify non-consecutive exposure to the same antibiotic during the same ICU stay. For example, a patient is exposed to 24h of PTZ, then switch to ceftazidime for 6 days to complete his treatment. Two weeks later, he develops a new infection and is re-exposed to PTZ for a total of 5 days. Allowing to capture each day of exposure, even if short in duration, allow us to increase our statistical power to identify small effect size (as the relative increased risk of AKI attributed to that exposure). We also included the duration of treatment as an adjustment co-variate in the multivariate analysis.

In addition, as suggested, we performed a re-analysis of the dataset while excluding patients receiving (in total, during the entire ICU stay) less than 72h of PTZ, Vancomycin or combination of both. As shown below (highlighted in yellow), no significant difference in major outcome results could be seen. These interesting results were included in the Suppl. Material. 

Q: It is unusual for patients who are on treatment for both P. aeruginosa and MRSA to not be prescribed either Piptaz or vancomycin. In my experience, if there is a concern re nephrotoxicity with combination vancomycin and piperacillin/tazobactam, patients are usually treated with cefepime or meropenem instead of Piptaz. I would be more interested in rates of AKI with cefepime/vancomycin or meropenem/vancomycin compared to Piptaz/vancomycin.

I like the included tables. However, I feel some of the most useful data is buried in the supplementary section, particularly Figure S1 and Tables S1 and S2.

R: We agree with you that, since the last 5-6 years, clinicians are more aware of the potential risk of nephrotoxicity with the PTZ-vancomycin combination. However, data from this study come from a cohort of patients hospitalised between 2001 and 2012. The potential risk of prescription bias (notoriety bias) could have been less important than if the study was performed on a more recent cohort. Nevertheless, if clinicians were aware of the potential risk of nephrotoxicity and modified their choice of antibiotics based on patients’ relative risk to develop AKI, it should lead toward the null hypothesis, as the overall risk of AKI should have been minimized as high-risk patients may receive non-PTZ-vancomycin combinations. Instead, we found a significant association between the risk of AKI and exposure to PTZ alone or in combination with vancomycin.

Regarding the usefulness of Tables and Figures integrated in the Suppl. Materials, we initially decided to minimize the amount of data showed in the main article to the essential. However, we agree with reviewer #1 that the Figure S1 is highly relevant, especially as it described the Odds associated with all co-variates incorporated into the adjusted analysis used for the primary outcome. As suggested, we decided to transfer that figure into the main text. 

Q: The discussion makes a good point about changes in indications for dialysis in ICU over the study timeframe.

Completely agree that the findings of this study add yet more weight to the requirement for prospective RCTs on this issue rather than further retrospective data

R: Thank you very much.

Reviewer #2: 

Q: Thank you for the opportunity to review, "Risk of Acute Kidney Injury associated with Anti-pseudomonal and Anti-MRSA Antibiotic Strategies in Critically ill Patients."

This is a retrospective cohort study of data from the MIMIC-III database including 18,510 encounters among 15,673 individual adult patients admitted to a single academic center’s ICUs over an 11-year period (2001-2012). The authors analyze the association between treatment with Vancomycin vs another anti-MRSA antibiotic, Piperacillin-Tazobactam (PTZ) vs another anti-Pseudomonal antibiotic, and combination therapy with Vancomycin-PTZ vs any other anti-MRSA-non-PTZ anti-Pseudomonal antibiotic combination with new or worsening acute kidney injury (AKI) within 7 days, according to creatinine-based KDIGO criteria. Secondarily, the authors analyze the association between the above antibiotics and new-onset renal replacement therapy within 7- and 30-days and changes in serum creatinine and serum urea among all study participants. The authors conclude that among ICU patients who received anti-Pseudomonal and anti-MRSA antibiotics, exposure to PTZ and vancomycin, individually or in combination, had an increased risk of new or worsening AKI within one week.

This is a well-performed research study that follows the STROBE criteria and adds to an existing body of evidence regarding the renal risks of exposure to various nephrotoxic medications in critically ill patients.

R: We would like to thank you for reviewing our work. 

Major revision suggestion:

My only major criticism is the authors’ decision to not perform propensity score matching in their retrospective analysis of this observational dataset. Many recent well-performed large-scale retrospective cohort studies seeking to elucidate the association between various interventions and acute kidney injury have used this statistical tool to good effect and have been published in PLoS One (PMID 30142190) and other journals (PMID 31952871). The authors should consider whether this would strengthen their analysis by accounting for the numerous patient characteristics that may contribute to clinicians’ decision to choose one antibiotic agent over another in critically ill patients.

R: Thank you for sharing these two interesting studies. As mentioned above (see response to the Editor), Propensity score matching is a powerful tool to adjust for co-variates and potential confounders. However, the use of a GEE and its repeated measures design did not allow us to consider a PS matching as an option. In addition, the size of the overall cohort, with more than 169 000 observation days, and the number of outcome events makes the usefulness of incorporating a Propensity Score as a co-variate much less clear1, especially as 15 relevant variables have already been included into the multivariable model. 

1 Elze, M.C. et al. J Am Coll Cardiol. 2017;69(3):345-57

Minor revision suggestions include:

Q: - Page 5, line 11: Missing closing parenthesis after “aztreonam”

R: Done

Q: - Page 14, line 1: Replace “but” with “except in” to make it clearer that these two sub-groups had different outcomes measured. As currently written, this is unclear.

R: Thank you for the suggestion. Done. 

Q: - Page 14, Discussion section: The authors might consider discussing whether their sub-group analysis demonstrating that, among patients with confirmed Pseudomonal infection, administration of PTZ was not associated with worsening AKI might encourage clinicians to more carefully consider which patients are at higher risk for Pseudomonas to help determine when PTZ may indeed be the optimal choice of anti-Pseudomonal antibiotic if it does not have a significant association with new or worsening AKI for patients who end up having a Pseudomonal infection

R: This is a really interesting suggestion, especially as severe confirmed pseudomonal infections are at high risk of therapeutic failure, and some guidelines now recommend dual anti-pseudomonal coverage – where PTZ remains an agent of choice. We added the following lines to the main text (discussion): 

“Severe pseudomonal infections are associated with higher risk of therapeutic failure. Some guidelines have therefore recommended the use of double anti-pseudomonal coverage in high-risk patients, such as ventilator-associated pneumonia(23). This study wasn’t designed to evaluate the additional risk of AKI in patients requiring dual anti-pseudomonal coverage. However, as shown in Figure 2 (and Table S5, Suppl. Material), among patients with confirmed pseudomonas infection, totalising 16,444 observation days, there was no increased risk of new or worsening AKI within 7 days associated with PTZ exposure. Therefore, in high-risk patients with known pseudomonas infection, PTZ may still represent an appropriate choice a priori.”

Q: - Page 17, line 17-19: Are the authors able to determine how often their studied antibiotics were prescribed but not administered? One might assume this would be a very small number and, if the authors are able to report how frequently this occurred, the reader would be better able to determine how much this inherent limitation of a database study should inform their conclusions.

R: This is a major limitation of observational studies using retrospective database. It is not possible to quantify the difference between prescribed drugs and administrated drugs in the MIMIC-III database. 

We modified the discussion section to further report this limitation:

“[…] Also, data on medication administration were based on prescriptions rather than on administered drugs, which is a limitation of such retrospective studies.”

Reviewer #3: 

This article examines the association of piperacillin-tazobactam with acute kidney injury (AKI) in a large single-center cohort of ICU patients. As stated by the authors, the concept is not new but the current study provides a more granular analysis of this association by controlling for common risk factors for AKI n ICU patients. Few comments and queries, if I may.

Q: 1. The finding that PTZ-vancomycin combination was more nephrotoxic than PTZ alone but not any more nephrotoxic than vanc alone is interesting and should be discussed further. How do the authors explain this finding? Does this suggest that when patients develop AKI while on this combination, the relative contribution of vancomycin is higher than PTZ? Does this finding suggest that for patients who--- for whatever reason--- are thought to require vancomycin as a preferred agent, the addition of PTZ should not expose them to higher risk of nephrotoxicity than vanc alone? Similarly, for those who require PTZ as the drug of choice (eg, when risk of cefepime-induced encephalopathy is considerable), the addition of vanc should be avoided unless it also is considered to be the drug of choice?

R: First, thank you for having reviewed our work. Regarding your question referring to the results presented in Table 4. Yes, we have the same interpretation as you: these results showed that the intrinsic nephrotoxicity of vancomycin is probably higher than the nephrotoxicity associated with PTZ exposure individually (which is compatible with previous data). In the univariate analysis (data not showed), the risk of developing the AKI outcome within 7 days occurred in 33.3% of observations with vancomycin only, in 34.9% of observations with the PTZ-Vancomycin combination and in 27.2% of observation with PTZ only. Therefore, once adjusted for the clinical model, no difference was observed between the vancomycin only and PTZ-vanco combination, but a significant difference was found when comparing PTZ-vanco to PTZ only. Here, we can consider that the major trigger of nephrotoxicity is probably the vancomycin itself. 

However, when comparing PTZ to another anti-pseudomonal agent individual, as shown in Table 3, or pooled altogether (Table 2) there was an increased risk of AKI when exposed to PTZ compared to all other agents, except for aminoglycosides, where the risk of AKI was higher in the latter. 

Thank for reporting this. We adapted the results section of the manuscript to improve that interpretation:

- Adding these event frequencies into the result section

Regarding the last question, results from an observational/retrospective study like this, despite the strong association observed, should be considered with caution. Until the elaboration of definitive prospective trials, no recommendation regarding the optimal MRSA/anti-pseudomonal combination can be elaborated. It is true that the risk of AKI should be part of equation when prescribing antibiotics, but other major aspects should also be integrated: cost, risk of antimicrobial resistance, local availability, etc.

Q: 2. Is there data examining 24, 48 or 72 hrs of the combination vs longer duration as relates to risk of AKI? Please discuss how the results of this study may be used (or cannot be used) to determine if there is a “safe” duration of PTZ-vancomycin combination beyond which the risk of AKI increases significantly.

R: This is a good question. The repeated measures design of the GEE analysis performed did not allow us to adequately evaluate the cumulative risk of AKI for each consecutive day of exposure, but instead for each additional observation day clustered at the patient’s level. Therefore, we cannot determine at which treatment duration, should we stop the PTZ-vancomycin combination to minimize the risk of AKI (time-to-AKI analysis). 

Others…

Q: 1. Abstract, line 17-18. Should be a complete sentence.

R: Done

2. Page 5, line 16. Should be “was”

R: Thank you. Done.

3. Page 6, line 13. Active bacteremia was defined as “any positive blood culture…” How were contaminants handled? Consider using the term “true bacteremia” with explanation of excluding contaminants based on clinical grounds.

R: It is not possible to identify true bacteremia from contaminants, as the MIMIC-III database doesn’t mention the clinical relevance of such “positive blood culture”. However, we agree with you that this limitation should be mentioned. We therefore added the following line in the discussion section: 

“Any positive blood culture was considered an active bacteremia as the MIMIC-III database did not allow to identify contaminant from true bacteremia”

4. Page 11, Table 2. Consider an alternative to “Intervention” column heading since this is a retrospective (not prospective) study. ? Investigated drugs

R: Thank you for this observation. We changed all columns heading accordingly. 

Thank you!

Reviewer #4: 

The manuscript examined the risk of AKI associated with anti-pseudomonal and anti-MRSA antibiotic strategies in critically ill patients. Here are some comments for consideration:

Q: 1. Page 3, line 6. May want to further expand on relationship between broad spectrum antibiotics and MDR organisms.

R: Thank you. We added the following lines to the introduction section:

“On the other side, inappropriate use of antibiotics is associated with the development of multidrug-resistant organism. The choice of the empirical antibiotic regimen should be individualised according to various factors such as local resistance rate, previous patient’s infections as well as the suspected site of infection(3).”

Q: 2. Page 3, line 11. Would want to update reference to the new Campaign recommendations for 2021.

R: Thank you for this observation. We updated the reference according to these new recommendations (were published after the initial submission). 

Q: 3. Page 3, line 21. The study by Blevins A. AAC; 2019; 63: e02658.may be useful to include as that study examined the rates of AKI with different severity in the ICU. The study by Contejean A. JAC 2021; 76: 1311 would also be useful in the discussion on AKI with vancomycin/piperacillin/tazobactam combination.

R: The study by Blevins et Al. is already cited in this manuscript. Results from this study, including the odds ratio of developing AKI for patients admitted in ICU is reported in the Discussion section. 

Regarding the second reference (Contejean et al.), it is a very interesting study using the WHO database of individual case safety reports that also confirmed the excess of AKI associated with the PTZ-vancomycin combination. We would like to thank you for this reference that we included in the manuscript. 

Q: 4. Page 5, line 2. Is there a specific time frame for overlap of antibiotics?

R: Antibiotics need to be received for at least 24h to be included as an observation. For example, if a patient received PTZ alone for 3 days, then received vancomycin for 4 days. The patient is considered as receiving PTZ in monotherapy for 3 observation days, then the PTZ-vanco combination for 4 observation days (considering these observations as cluster). 

Q: 5. Page 5, Observations and endpoints. Would be useful to define severe and non-severe AKI as this was discussed as a limitations of previous studies.

R: It is an interesting suggestion, as we consider one of the power of our work is to report that severity. As suggested;

- In the Method section, we mentioned that new-onset KRT within 7 and 30 days served as proxy for severe AKI. 

- In the Results section, we added the following line: “Stage 3 AKI or KRT initiation within 7 days occurred in 2,835 (21%) of patients exposed to anti-pseudomonas, in 3,064 (19%) exposed to anti-MRSA and in 2,586 (24%) exposed to both coverage for at least 24 hours”

6. Page 5, line 21. Can further clarify day 7 observation period, is that after receipt of combination antibiotics or any antibiotics?

R: The 7 days observation period is the next 7 days following the receipt of one of the antibiotics of interest. Ex: If a patient is exposed to Ceftazidime for 3 consecutive days (D1 to D3). The observation period is the day D1 to D7 for the observation day #1, D2 to D8 for the observation day #2 and D3 to D9 for the observation day #3. To further illustrate this important concept, we added a new Figure in the Suppl. Material reporting the timeline of follow-up for a hypothetic patient.

7. Page 6, covariates. Was patients in shock included in the study? If so, where they on vasopressors as this was considered a major risk factor for Aki development.

R: Yes, vasopressors (at any dose) were administered in 31% of all patients included. This variable was associated with an increased risk of AKI in the univariate analysis (Figure 1) and was therefore included as an adjustment co-variate in the multivariate model. 

8. Page 7, line 10. Were comorbidities like shock, other nephrotoxins, and amount of fluids administered included in the covariate?

R: Shock could be defined with the presence of vasopressors (co-variate) as well as using the SOFA Score (also a co-variate in the multivariate model). The amount of fluid administered was not included in the model, nor the fluid balance because the analysis used a repeated measures design (GEE), which means an exposure to a single antibiotic can occurs multiple times during an entire ICU stay (ex: Day 1 to 3, then D31 to 37, then D39 to 47). Fluid balance and volume of fluid administered cannot be easily included in this model, as the same patient is not followed continuously from admission. 

We therefore add a new limitation to the discussion section: “The fluid balance could not be included due to the statistical design with repeated measures” As the concept of under-resuscitation and high positive fluid balance are both associate with increased risk of AKI.

Q: 9. Page 7, line 17. What is defined as toxic vancomycin level?

R: More than 20 mg/L. We adapted the text accordingly. 

Q: 10. Page 8, table 1. Is there a breakdown of what stage of CKD for the patient population?

R: Yes. We adapted the Table 1 accordingly.

Q: 11. Page 8, table 1. Is the mean time to treatment and mean time to AKI available?

R: No. This variable has not been generated using the original MIMIC III dataset. However, as shown in Table 1, most patients received at least one of the antibiotics of interest within the first 48h (74%). 

Q: 12. Page 10. Is the data for AKI by severity available?

R: The primary endpoint is new onset AKI or AKI progression within 7 days. As some patients had already AKI criteria at the time of antibiotics initiation, we considered this primary endpoint definition more clinically relevant, especially in a population at high risk of AKI (from baseline) as in the ICU. However, when considering patients individually, it is possible to identify the maximum AKI stage achieved during the 7days follow-up (Table S1). As suggested above, we included these non-adjusted results for severe AKI events into the Results Section: “Stage 3 AKI or KRT initiation within 7 days occurred in 2,835 (21%) of patients exposed to anti-pseudomonas, in 3,064 (19%) exposed to anti-MRSA and in 2,586 (24%) exposed to both coverage for at least 24 hours (Table S1 and Table S2 in the Suppl. Material)”. 

Q: 13. Page 14, line 6. This is referring to Table S5 rather than S6.

R: Thank you for noticing this typo. 

14. Page 14, line 11. Can expand on this statement a little more.

R: The text was adapted according to the suggestion in order to improve comprehension of this concept: 

- In the Result section, we added: “However, in patients who progressed to stage 2 or 3 AKI within 7 days, exposure to the PTZ-vancomycin combination was associated with a higher increase in serum creatinine than other anti-pseudomonas/anti-MRSA regimens (5.6% [95% CI, 3.1-8.2%, p<0.001]), with no significant difference in creatinine elevation in patients with limited stage 1 AKI. Instead, in that group with non-severe AKI, patients exposed to PTZ, with or without concomitant vancomycin, achieved the AKI creatinine elevation criteria despite a reduction in serum urea within 72h.”

- In the Discussion section: “We also showed that, when exposed to PTZ as opposed to other regimens, patients with non-severe AKI achieved the stage 1 AKI criteria (based on serum creatinine) despite a relative reduction in serum urea, which could be partially attributed to a inhibition of creatinine tubular secretion. However, a moderate elevation in serum creatinine solely due to inhibition of tubular secretion is unlikely to translate into an increased risk of severe AKI and KRT use”

Q: 15. Page 14, discussion. Can discuss more on the overall incidence rate of AKI in this study compare to other studies. Can also discuss about the difference in outcome when observing the rates of AKI in patient receiving piperacillin vs those who received vancomycin.

R: Thank you for this suggestion. We added the following line to the Discussion section:

“In addition, the proportion of patients who progressed to the primary endpoint of new or worsening AKI (57.0%) was slightly higher than usually reported in previous ICU cohorts(23), which might be due to the use of KDIGO criteria and selection of relatively sick patients with active infection required broad-spectrum antibiotics.”

Q: 16. Page 17, line 21. Can comment more about the limitation of using KDIGO-AKI criteria vs. the other definitions.

R: Thank you very much for the suggestion. Modifications were made accordingly. We added: “(Using KDIGO-AKI criteria) which is sensitive to minor creatinine elevations and might have led to identification of AKI events with no clear clinical significance”.

---

## [Editor Report · Decision Letter 1]

8 Feb 2022

Risk of Acute Kidney Injury associated with Anti-pseudomonal and Anti-MRSA Antibiotic Strategies in Critically ill Patients

PONE-D-21-34271R1

Dear Dr. Cote,

We’re pleased to inform you that your manuscript has been judged scientifically suitable for publication and will be formally accepted for publication once it meets all outstanding technical requirements.

Kind regards,

Eili Y. Klein, PhD

Academic Editor

PLOS ONE
---

## [Editor Report · Acceptance letter]

1 Mar 2022

PONE-D-21-34271R1 

Risk of Acute Kidney Injury associated with Anti-pseudomonal and Anti-MRSA Antibiotic Strategies in Critically ill Patients 

Dear Dr. Côté:

I'm pleased to inform you that your manuscript has been deemed suitable for publication in PLOS ONE. Congratulations! Your manuscript is now with our production department. 

Kind regards, 

on behalf of

Dr. Eili Y. Klein 

Academic Editor

PLOS ONE